# Do soil health indicators predict carbon and nitrogen functional stability under drought and heat?

Patricia Lazicki[1][¤a]*, Jaehoon Lee[1], Alemu Mengistu[2], Sindhu Jagadamma[1]*

**1** Department of Biosystems Engineering and Soil Science, University of Tennessee, Knoxville, Tennessee, United States of America, **2** USDA-ARS Crop Genetics Research Unit, Jackson, Tennessee, United States of America

¤a Permanent address: University of California Cooperative Extension, Capitol Corridor, Woodland, California, United States of America

* palazicki@ucanr.edu; sjagada1@utk.edu

## Abstract

Healthier soils are often assumed to retain function better under climate stress. However, links between common soil health indicators and soil functional resilience to stress are elusive. Our goal was to link soil health status with stress response by quantifying the multifunctional carbon (C) and nitrogen (N) cycling response of soils under different management (forest, conventional, or organic) to drought or combined drought and heat stress. We monitored several C and N cycling functions during a 28-d stress and 28-d recovery period and calculated resistance and resilience indices for each function to create multifunctional C and N cycling indices. We related these indices to baseline soil health properties and microbial community characteristics. Traditional soil health indicators (e.g., total organic C, and microbial biomass and activity) were closely associated with N cycle resilience to drought stress. Indicators that distinguished forest from arable soils (e.g., low pH and chemical fertility, and high porosity, relative abundance of Basidiomycetes, and high C to N ratio) were generally positively related to drought resistance but negatively related to resilience. Bacterial and fungal community diversity were unrelated to either resistance or resilience for either cycle. Adding heat to drought created a strong N cycle stress which affected all sites similarly, and soil baseline properties were not related to either resistance or resilience. For both C and N cycles, stress type was the major determinant of resistance while management was the major determinant of resilience. Our results show that response to drought stress differs depending on the temperature at which it occurs, and that pH, chemical fertility, and SOM are all important components in stress response but affect C and N cycles differently.

**Data availability statement:** All data files are available from the Dryad database (DOI: https://doi.org/10.5061/dryad.1c59zw45q).

**Funding:** USDA ARS (grant number 58-6066-8-043); Tennessee Department of Agriculture (grant number A20-0337).

**Competing interests:** The authors have declared that no competing interests exist.

## 1. Introduction

Drought, heat waves, and intense rainfall events are predicted to become more common with climate change. These extreme climate events can all be stressors on the bacteria, fungi, and archaea that drive important soil functions such as carbon (C) and nitrogen (N) cycling [1,2]. Management which improves soil health (defined by the U.S. Department of Agriculture Natural Resource Conservation Service as "the continued capacity of a soil to function as a vital living ecosystem that sustains plants, animals, and humans") is often assumed to also make soil function more resistant (RS) and resilient (RL) to stresses caused by climate fluctuations. That is, important microbially-driven functions in "healthy" soils are assumed to change less during stress (RS) and to more rapidly and closely resemble those of non-stressed soils when conditions return to normal (RL) [3]. Indeed, stability to disturbance has been proposed as a key unifying indicator of soil health [3, 4, 5]. However, soil health assessment schemes, which normally include soil organic matter (SOM) and its various fractions, available nutrients, physical properties, and indices of soil biological activity such as respiration or potential enzyme activities, generally do not include how these pools and functions respond to climate stress (e.g., [6,7]). Furthermore, the relationship between soil health and functional stability is likely not straightforward.

The relationship of soil health indicators and functional stability under stress likely depends on the type and severity of the stress in question and the initial microbial community [8]. Ecologically, a healthier system is one with more internal self-regulation, in which greater disturbance is needed to shift it from a desired state, and fewer external inputs are needed to maintain it in such a state [9]. Many properties measured by common soil health assessment methods, such as SOM and structural properties, would theoretically underly such regulation [6]. For example, soils with high porosity and water holding capacity could lose water or conduct heat more gradually, reducing microbial mortality under heat or drought stress compared with a poorly structured soil under comparable stress [10]. Fertile, high-SOM soils could support faster community regrowth after a comparable mortality event than infertile, low-SOM soils [8]. However, "healthy" soils may also have microbial communities which are poorly adapted to stress, and therefore are more affected by stressful conditions than communities from "less healthy" soils [11]. For example, microbial communities from poorly structured soils may contain more members adapted to drought or low-oxygen conditions than communities from well-structured soils, and so may lose less function if the same drought or flooding stress is imposed on both soils [12,13].

As different microbial community members will be differentially affected by a stress, the different processes they perform and the products of those processes will also respond differently. An approach for synthesizing the responses of multiple facets of a larger ecosystem process is to combine them in a multifunctionality index (e.g., [14,15]). This approach has long been used in ecology to study the links between organisms and ecosystem function, as it provides a means of summarizing the aggregate functional change of an ecosystem in response to a change that

simultaneously affects all members of the community [16]. More recently, the approach has been combined with RS and RL indices to relate the stress behavior of soil processes to soil and microbial community characteristics [17,18]. However, we are not aware of any work which test the relationship of common soil health indicators with multifunctional stability of soil functions to climate stress.

In this study, our objective was to answer the question: "Which management-sensitive soil health indicators and microbial community characteristics are most associated with RS and RL to a simulated climate stress?" To that end, we tested RS and RL of multifunctional C and N cycling in differently managed soils to single (drought) and compounded (drought + heat) disturbances. We then performed a microbial community analysis and an extensive soil health assessment and related soil and community characteristics to multifunctional RS and RL indices.

We hypothesized that 1) Adding heat to drought would decrease RS and RL of all measured C and N cycle metrics at all sites [11], 2) RS would increase with soil porosity and soil C, as such soils will likely lose water and transmit heat more slowly, due to better water holding capacity and improved thermal resistivity [10,19], 3) High chemical fertility would be associated with high RL, as more available nutrients will support faster microbial regrowth [8], and 4) Higher bacterial and fungal diversity would be associated with high RS and RL, as increased functional redundancy increases the likelihood of organisms ability to perform any given function under both stress and recovery conditions [18].

## 2. Methods

An experimental overview is given in Fig. 1.

### 2.1 Site management history

All sites were located in Robertson County, Tennessee, USA. Sites were accessed with grower permission. The climate is humid subtropical with a Köppen climate classification of Cfa. The normal annual precipitation (1991–2021) is 1380 mm and normal average annual temperature of 15°C (National Oceanic and Atmospheric Administration, 2022). The CONV field (36°35'N, 86°36'W) was in corn-soybean/wheat rotation and had been under no-till management for 12 yr. A grass-legume cover crop was grown approximately every 2–3 yr. The site received synthetic herbicides, pesticides, and fertility. The ORG field (36°35'N 86°41'W) was in a 6-yr corn-soybean-pasture rotation, with a multi-species cover crop

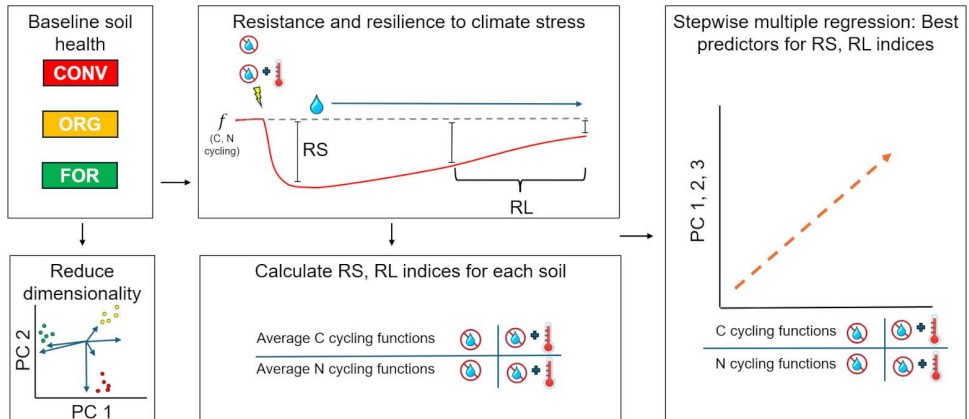

**Fig 1. Experimental schematic overview.** Soils are taken from CONV = Conventional; ORG = Organic; FOR=Forest sites. Baseline soil health indicators and microbial communities were measured, and a principal components analysis performed to reduce dimensionality to uncorrelated variables differentiating sites (PCs). Cores were subjected to either heat or heat + drought stress and then incubated under non-stressed conditions. Several assays of C and N cycling functions performed at maximum stress (resistance, RS) and during recovery (resilience, RL). RS and RL indices were calculated compared to a non-stressed control. Stepwise multiple regression was used to relate stress indices for both cycles to baseline soil PCs.

grown after arable crops, for at least 20 yr. Poultry manure compost was applied prior to corn years. The site had been under no-till management for the1.5 yr prior to sampling, with the cover crop terminated by grazing. Prior to that cover crops and compost were incorporated by discing. The FOR site (36°35'"N 86°41'W) was a strip of unmanaged oak-hickory secondary forest adjoining the ORG field.

## 2.2 Soil sampling

Undisturbed soil cores (10 cm depth, 5.08 cm diameter) were obtained from each of the three sites in early August 2021. Fifty-five cores were taken per site (10 baseline + 5 replicates x 3 treatments x 3 destructive sampling dates). The ORG and CONV soils were under corn in the early reproductive phase. Soils at all sites were mapped as Pembroke silt loam (Fine-silty, mixed, active, mesic Mollic Paleudalfs; Soil Survey Staff, 2014). At each site, 25 additional samples were taken with a 2-cm diameter probe to a depth of 10 cm to obtain a representative bulk sample for analysis of additional baseline data. This sample was composited and homogenized. A portion was immediately placed on dry ice and transported to lab. Subsamples were stored at −80 °C for microbial community quantification via 16s and ITS analyses and at −20 °C for enzyme analysis. Soil microbial DNA was extracted from five 0.25-g replicate subsamples of the frozen composite sample collected at each site using the Qiagen DNeasy PowerSoil kit, according to the manufacturer's instructions (Qiagen, Valencia, CA, United States). Enzyme analyses were performed using the fluorescence microplate method (Table 1). Details of DNA extraction, community analysis, and enzyme assays are given in the S1 File. The remainder of the composite sample was spread in a thin layer to air-dry and sieved to <2 mm for texture, water-stable aggregate (WSA), and carbon fraction analysis (Table 1).

## 2.3 Baseline analyses

Ten undisturbed cores from each site were randomly allocated for baseline analyses (Table 1). Details of all analyses are given in the S1 File. Bulk density and thermal properties were measured on undisturbed cores. Cores were then sieved to <4 mm and immediately analyzed for ammonium, nitrate, microbial biomass C and N (MBC, MBN), and the C and net N mineralization during a 7-d incubation at 60% water holding capacity (WHC). The remaining soil was dried in a thin layer, sieved to 2 mm, and analyzed for several soil fertility and soil health properties according to standard methods (Table 1).

## 2.4 Incubation

A 56-d aerobic incubation was performed on undisturbed cores to assess C and N cycling RS and RL to drought (DRT) or combined heat and drought (H + D) stress. Full details of the incubation are given in [20]. Briefly, all cores were brought to a water content of 0.3 g g$^{-1}$ dry soil. After a 7-d preincubation at 21 °C, cores for DRT and H + D treatments were allowed to dry naturally while reference (REF) cores were maintained at constant moisture. After 14 d of drying at 21 °C, H + D cores were removed to a 30 °C incubator. At day 28 of the incubation all cores were brought to a water content of 0.3 g g$^{-1}$ dry soil and maintained at constant moisture and 21 °C for 28 d. Cores (n = 5 per site and treatment) were destructively harvested just before rewetting for RS assessment, and 14 and 28 d after rewetting for RL assessment (Sect. 2.4.2).

 **2.4.1 Soil respiration measurement.** Soil respiration was measured on days 2, 5, 12, 15, 18, 21, 28, 30, 31, 32, 33, 36, 42, 49 and 56 of incubation. Five cores were randomly chosen from each site and treatment and placed in 2.37 L plastic jars with lids equipped with rubber septa for gas sampling, and incubated for 24 h at the same temperature as the current treatment (21 °C or 30 °C). Headspace air was analyzed for $CO_2$-C on a gas chromatograph (Shimadzu, Kyoto, Japan). Respiration rates for each day were estimated as the average 24-h respiration of the two closest sampling dates. The cumulative respiration was calculated as the sum of the estimated daily rates. On days 28, 42, and 56, the cores used for respiration sampling were immediately harvested.

**2.4.2 Carbon and nitrogen cycling functions.** Harvested cores were gently crumbled by hand and sieved to < 4 mm, and C and N assays were performed as described in detail [20]. Fresh sieved soil was analyzed for gravimetric water content (GWC), MBC, MBN, extractable organic C and N (EOC, EON), mineral N, and hydrolytic enzyme activity using the methods outlined in Table 1. A 7-d incubation was performed to assess functional stability to climate stress, using a short-term residue decomposition method adapted from [21]. Four 5-g subsamples of fresh soil were weighed into 40-ml glass vials. Finely ground multispecies cover crop residues (C:N ratio = 14:1) were mixed thoroughly with soil in two vials at a rate of 2.6 mg g$^{-1}$ dry soil (1000 mg C kg$^{-1}$ dry soil and 70 mg N kg$^{-1}$ dry soil). The cover crop mix included wheat (*Triticum aestivum*), white clover (*Trifolium repens*), and hairy vetch (*Vicia villosa*). The duplicate residue-added soils were adjusted to 60% WHC with deionized water and placed together uncovered into a 907-ml glass jar with an airtight lid fitted with a rubber septum for headspace sampling. The other two vials were treated similarly, except that no residue was added. Jars were incubated at 21 °C for 7 d. Jar headspace $CO_2$-C was measured after 24 h, 72 h and 7 d on a gas chromatograph (Shimadzu, Japan). Jars were aerated after each sampling event. The soil in each pair of vials was analyzed for MBC, MBN, and mineral N. Potential C mineralization from SOM was measured as the cumulative C evolved from the jar containing unamended soil. Apparent C and net N mineralization in response to residue addition was calculated as the difference in cumulative C respiration or mineral N between vials incubated with and without residue additions. Apparent MBC and MBN change in response to residue additions were calculated as the MBC and MBN difference between vials incubated with and without residue additions.

**Table 1. Baseline analysis of soil physical, biological, and chemical properties.**

| Analyses | Soil sample type (n = 5 for all) | Reference |
|---|---|---|
| **Physical** | | |
| Bulk density/ porosity | Undisturbed core | Linn and Doran, 1984 |
| Thermal properties | Undisturbed core | Haruna et al., 2017 |
| Gravimetric water content at sampling/ water holding capacity | Freshly sieved (<4 mm) cores | |
| Particle size distribution | Dried composite, sieved <2 mm) | Gee and Bauder, 1996 |
| Aggregate stability | Dried composite, sieved <2 mm | Kemper and Roseneau, 1996 |
| **Biological** | | |
| Bacterial and fungal diversity (Inverse Simpson index), richness (Chao 1 index) and evenness; relative abundance of major phyla | Fresh composite, stored at −80 °C | Callahan et al., 2017 |
| Microbial biomass C and N (Fumigation extraction) | Freshly sieved cores (<4 mm) | Horwath and Paul, 1996; Cabrera and Beare, 1993 |
| Potential enzyme activity (BG, LAP, NAG, XYL)† | Fresh composite, stored at −20 °C | Bell et al., 2013 |
| 7-d C and net N mineralization from fresh samples | Freshly sieved cores (<4 mm) | Lazicki et al., 2021 |
| 24-h respiration from rewet samples | Dried cores, sieved <2 mm | Lazicki et al., 2021 |
| **Chemical** | | |
| Soil organic C, total N, C:N ratio | Dried cores, sieved <2 mm | Nelson and Sommers, 1996 |
| Permanganate-oxidizable C | Dried cores, sieved <2 mm | Culman et al., 2022 |
| Particulate- and mineral-associated organic matter | Dried cores, sieved <2 mm | Cambardella and Elliot, 1992 |
| Extractable organic C and N | Freshly sieved cores (<4 mm) | Jones and Willet, 2006; Cabrera and Beare, 1993 |
| Mehlich 1-extractable nutrients (Ca, Mg, K, P, Zn) | Dried cores, sieved <2 mm | Mehlich, 1953 |
| pH (2:1 in de-ionized water) | Dried cores, sieved <2 mm | Thomas, 1996 |
| Ammonium and nitrate (summed as mineral N) | Freshly sieved cores (<4 mm) | Forster et al., 1978; Doane and Horwath, 2003 |

† BG = β-glucosidase; LAP = leucine aminopeptidase; N-acetyl-β-glucosaminidase (NAG), and β-xylosidase (XYL).

Functions were designated as related to C or N cycling. The nine N cycling functions were a) From the undisturbed cores: mineral N, the proportion of the mineral N measured which was in the form of $NH_4$, MBN, EON, and the potential activity of the enzymes leucine aminopeptidase (LAP) and N-acetyl-β-glucosaminidase (NAG) from undisturbed cores; b) From unamended, sieved soil during incubation: net N mineralized over 7 d and c) From residue-amended soil after 7 d incubation: additional mineral N and MBN, compared to unamended soils.

The ten C cycling functions were: a) From the undisturbed cores: cumulative total respiration per unit MBC, MBC, EOC, and the potential activity of the enzymes β-glucosidase (BG) and xylanase (XYL) b) From unamended soil during incubation: $CO_2$-C mineralized over 24 h and 7 d, $CO_2$-C mineralized over 24 h per unit MBC (q$CO_2$), and c) From residue-amended soil over 7 d incubation: additional MBC and $CO_2$-C measured, compared to unamended soils.

**2.4.3 Resistance and resilience calculations.** Resistance (RS) and resilience (RL) indices were calculated for each function using the method of Orwin and Wardle [22]. This method is commonly used in microbial response studies as it a) increases monotonically as RS or RL increase, b) gives equivalent values for a response of the same magnitude, whether it is higher or lower than that of the control, c) avoids infinite values and zero denominators and d) standardizes the recovery value to the initial disturbance [22]. Resistance (RS) was assessed for each variable as the difference between the stressed and the reference soil at the end of the disturbance (day 28), normalized to the value of the reference (unstressed) core as

$$RS = 1 - \left( \frac{2\,|D_0|}{(R_0 + |D_0|)} \right)$$
(1)

where $D_0$ is the difference between the value of each sample and the reference samples and $R_0$ is the value of the reference samples, measured at 28 d. As net N immobilization is expressed as a negative number, the N mineralized from the soil and the additional N mineralized from residue were normalized to the absolute value of $R_0$ in order to maintain the properties of the index (Orwin and Wardle, 2004). Resilience (RL) was calculated for each variable at 14 d and 28 d after rewetting as

$$RL = \left( \frac{2\,|D_0|}{(|D_0| + |D_x|)} \right) - 1$$
(2)

where $D_0$ is as defined in Eq 1 and $D_x$ is the difference between value for the disturbed soil and the average of the reference soils at sampling dates 14 and 28 d after rewetting. RL values for 14 d and 28 d after rewetting were averaged to give a single RL value for each variable.

Indices calculated for each function were averaged across functions designated as related to C- or N-cycling as noted above to create indices for C and N cycling multifunctionality [18].

## 2.5 Statistical analyses

Microbial community alpha diversity (fungal and bacterial evenness, Chao 1 richness and the inverse Simpson diversity index) for the baseline soils were calculated using the *vegan* package in R [23]. The Shannon and Simpson indices were also tested, but the inverse Simpson was chosen as giving the smallest within-group variability. Bacterial and fungal community difference among sites was assessed with the PERMANOVA test using the adonis2 function of the *vegan* package in R. Differences among sites' initial soil properties including microbial community alpha diversity measures and the abundance of major phyla were assessed by a one-way ANOVA in PROC GLIMMIX in SAS (SAS Corporation, Cary, North Carolina), with post-hoc means separation using Tukey's HSD. Assumptions were checked by Shapiro-Wilks' test

for normality and visual inspection of the residual plots, and values were log-transformed if needed to meet assumptions. Where such transformation was ineffective, differences within sites were assessed by Kruskal-Wallis' test in PROC NPAR-1WAY in SAS, with post-hoc means comparisons using the Dwass, Steel, Critchlow-Fligner Method. Differences between indices of C and N cycling RS and RL were assessed using two-way ANOVA in PROC GLIMMIX in SAS, with post-hoc means separation using Tukey's HSD. Site and Treatment were considered to be fixed effects and replicate to be random. Assumptions were tested as described above.

A principal components analysis (PCA) was performed using all baseline soil and microbial community variables (evenness, inverse Simpson and Chao diversity indices, and relative abundances of major fungal and bacterial phyla) using PROC Princomp in SAS. Major phyla were considered to be those with a relative abundance comprising at least 5% of the total bacterial or fungal population. The first four principal components were used as predictors in a stepwise multiple regression to test the relationships between baseline soil characteristics and C and N cycling RS and RL, using PROC REG in SAS. The entry threshold was set at $p < 0.15$. Assumptions were checked by visual inspection of the residuals.

## 3. Results and discussion

### 3.1  Differences in initial microbial communities and soil environment among land-use types

#### 3.1.1  Variations in microbial communities across land-use types.
For both bacteria and fungi, community composition varied more than alpha diversity. PERMANOVA analysis showed that bacterial communities differed significantly among sites ($p = 0.002$). Major phyla whose relative abundance differed significantly among sites were Planctomycetes (FOR>CONV=ORG; $p = 0.009$), Proteobacteria (FOR=ORG>CONV, $p = 0.003$), Firmicutes (ORG=FOR>CONV; $p = 0.01$), Bacteroidetes (ORG>CONV>FOR; $p < 0.001$), Actinobacteria and Gemmatimonadetes (CONV>ORG>FOR; $p < 0.001$ for both), Acidobacteria (CONV>ORG; FOR intermediate, $p = 0.02$), and Chloroflexii (CONV>ORG=FOR) (Supplemental information, S1 Figa. The only major phylum with similar abundance in all three sites was Verrucomicrobia. Bacterial evenness was significantly higher in the ORG site than in the other two ($F = 7.4$, $p = 0.01$). However, bacterial richness and diversity, as measured by the Chao 1 and Inverse Simpson indices, respectively, did not differ among sites. Fungal richness was significantly higher in the FOR site than the other two ($p = 0.002$), while diversity and evenness were similar (Supplementary S1 Figb). Fungal communities, like bacterial, differed significantly by site ($p = 0.002$). Major fungal phyla which differed significantly among sites were Basidiomycetes (FOR>CONV=ORG; $p < 0.001$), Ascomycetes (ORG=CONV>FOR, $p = 0.01$), Chytridiomycetes (CONV=ORG>FOR, $p < 0.001$), and Glomeromycetes (CONV>ORG>FOR, $p < 0.001$).

#### 3.1.2  Important differentiating site soil characteristics.
Although the soil at the three sites was classified as the same type, their texture differed slightly but significantly. The CONV soil was more clayey than the other two sites (CONV>ORG>FOR; $p < 0.05$). Clay percentages were 17, 14, and 10%, respectively. The FOR site had the most silt (FOR>ORG>CONV; $p < 0.05$), with concentrations of 81, 81, and 75%, respectively. FOR and CONV sites both had more sand than the ORG site ($p < 0.05$), with concentrations of 9, 8, and 6%, respectively. These differences in texture suggest that although mapped as the same type, non-management-induced edaphic factors may also have contributed to some of the differences noted below.

Most baseline soil health metrics differed significantly among sites. Mean values of all soil properties are given in Supplementary Table S1. Principal components analysis was used to reduce the data dimensionality to a few uncorrelated variables that captured most of the variability in the dataset, and to detect variables which were particularly influential in differentiating the sites. Mean values of all variables and their loadings for the first four principal components are given in Supplemental Tables S1 and S2. The first principal component (Prin 1; "Arableness") separated the arable sites (ORG and CONV) from the FOR site (Fig. 2). The most influential variables were pH, extractable Ca and Mg, clay concentration, and the relative abundance of Gemmatimonadetes (high in arable sites); and porosity, thermal resistivity, SOM C:N ratio,

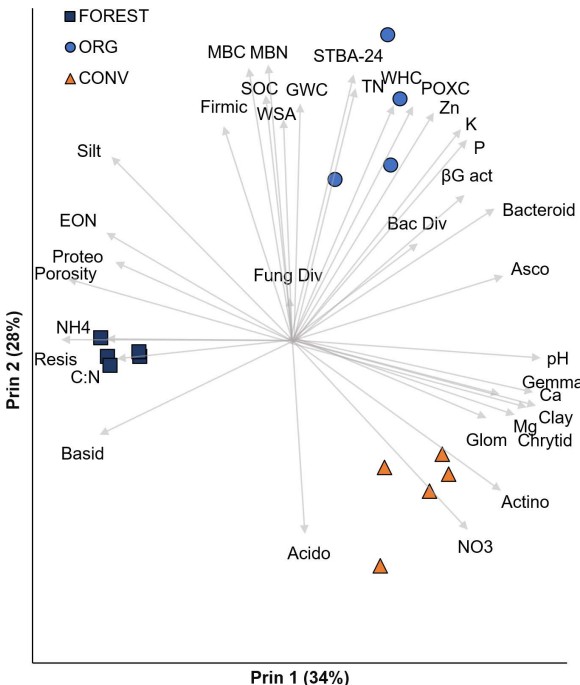

**Fig 2. Principal component analysis of soil properties. and microbial communities.** Some indicators not shown for clarity. Relative abundance of bacterial phyla Proteobacteria (Proteo), Firmicutes (Firmic), Bacteroidetes (Bacteroid), Gemmatimonadetes (Gemma), Actinomycetes (Actino) and Acidobacteria (Acido) and fungal phyla Basidiomycetes (Basid), Ascomycetes (Asco), Chrytidiomycetes (Chrytid), and Glomeromycetes (Glom); soil properties microbial biomass C and N (MBC, MBN), soil organic C (SOC), total N (TN), and C:N ratio (C:N), water-stable aggregates (WSA), gravimetric water content (GWC), 24-h rewet respiration (STBA-24), water-holding capacity (WHC), permanganate oxidizable C (POXC), Mehlich 1-extractable Zn, K, P, Ca, and Mg, pH, extractable organic N (EON), bacterial and fungal diversity (Bac & Fun Div), potential β-glucosidase activity (βG act).

and relative abundance of Basidiomycetes (high in the FOR site). The second principal component (Prin 2; "Soil Health") separated the sites in order of SOC concentration (ORG>FOR>CONV). Variables associated with Prin 2 were those commonly used as indicators for agricultural soil health, including MBC and MBN, biological activity as measured by 24-h respiration from rewet soil, permanganate-oxidizable carbon (POXC), WHC, and water-stable aggregates. The SOM-poor CONV site was differentiated by high $NO_3$-N concentration and a high relative abundance of Acidobacteria. The third principal component (Prin 3, "alpha diversity", accounting for 11% of dataset variability) loaded highly for bacterial species richness, bacterial and fungal diversity, fungal evenness, and negatively for bacterial evenness, while the fourth (6%) loaded highly for potential xylanase activity (Supplementary Table S2). Neither Prin 3 nor Prin 4 differed among sites.

### 3.2 Resistance (RS) and resilience (RL) of C and N cycling varied by site and stress type

Results of individual assays from the incubation experiment have been published in a companion paper [20]. As we predicted with Hypothesis 1, when heat was added to drought it significantly decreased both C and N cycle RS (p = 0.02 and P = 0.0007, respectively, for the main effect of treatment; Fig 3a– 3b). Nitrogen cycle RS in the FOR cores was especially impacted by the addition of heat: out of the three sites FOR had the highest RS to DRT but the lowest to D + H (Fig 3b). Both C and N cycles under DRT site RS decreased numerically as FOR<ORG<CONV (Figs 3a and 3b). However, the difference was not significant between individual sites.

While RS varied more by stress type, RL varied more by land use practice (p < 0.0001 for site, both C and N cycles, Figs 3c and 3d). Contrary to our hypothesis, carbon cycle RL was not significantly affected by stress type (Fig 3c), while

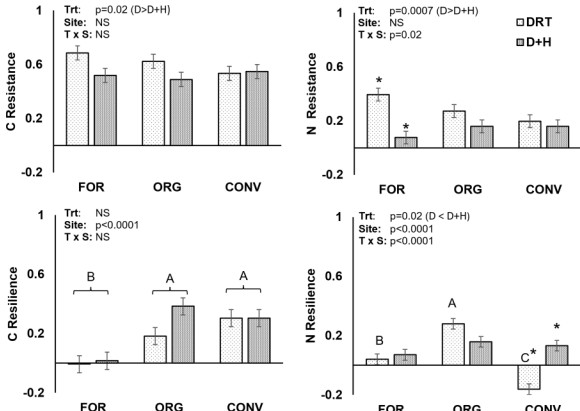

**Fig 3. Resistance (a,b) and resilience (c,d) of C (a,c) and N (b,d) cycling functions in response to heat and drought stress.** Bars represent the standard error of the mean (n = 5). Different lowercase letters represent significant differences between site * treatment combinations (p < 0.05, using Tukey's HSD test for post-hoc multiple comparisons).

stress effects on N cycle RL varied sharply by site (Fig 3d). The FOR and ORG sites displayed similar N cycle RL to both stresses. However, the CONV site showed a negative RL to DRT alone, meaning that N cycle metrics in the stressed treatments differed more from the control at the end of the recovery period than they did during maximum stress. Overall, ORG site RL was always equal to or greater than that of the other sites for both stresses and cycles. Conversely, the FOR site had essentially no RL to either stress for either cycle. That is, differences between stressed and control soils changed little during the recovery period.

Studies from Mediterranean climates have found that extreme heat waves (e.g., drought accompanied by temperatures >40 °C) reduce RS compared to drought alone, likely through increased microbial mortality [1,11]. Conversely, other studies have found that mild warming can mitigate the effects of drought, possibly by increasing metabolic process rates [24, 25, 26]. Although the temperature used in our study (30 °C) was unlikely to cause significant mortality [2], our results are more in line with the Mediterranean studies conducted under more severe stresses. A possible explanation is that soils with less history of drought stress have a larger response to drought than more adapted soils [27,28]. This provides evidence for the theory that stress response is affected by microbial community adaptation as well as to the magnitude of the stress and the soil environment [29]. This mechanism likely explains the high RS to DRT but low RS to D + H in the shaded, litter-covered FOR site. Similarly, Guillot et al. [11] measured the lowest microbial biomass RS to D + H in soils collected from under the tree row in an agroforestry experiment, while the highest RS to D + H was measured in soils collected from the cropped inter-row. Additionally, analysis of a network of climate change studies across the US found increased proteolytic activity in temperate forests under warming and drought [30]. Such increased activity, combined with a lower microbial biomass to immobilize the products, may also have contributed to the low FOR site RS to D + H.

Multifunctional C cycling was more resistant to both stresses than that of N (Figs 3a, 3b). This is consistent with findings from both laboratory and in situ climate manipulations, that drought stress affects N cycling processes more strongly than C cycling processes [31, 32, 33]. In a similar experiment, Guillot et al. [11] also observed a greater decrease in MBN than MBC under stress, particularly for D + H, resulting in an altered microbial community stoichiometry. Microbial biomass N was also disproportionately affected in our study, accompanied by a greater increase in potential N mineralization than C mineralization [20]. This suggests that microbial stoichiometry changes are partly responsible for the low multifunctional N cycle RS compared to C cycle RS.

With the potential exception of C cycle response to DRT, our data did not show any evidence of a tradeoff between RS and RL. Many studies find RS is inversely related to RL [8,11,26]. This is partly due to the widespread use of an

impact-normalized index [22], which puts the RS in the denominator and so can artificially inflate the RL when RS is low. Additionally, RL depends on the presence of available resources to support recovery of the organisms present [8.11]. Physiological alterations are energetically expensive and the organisms that use them are less likely to die or go dormant [26]. Thus, compared with susceptible populations, resistant organisms are less likely to release nutrients by lysing during stress and may also deplete nutrient pools more [26,34]. So, more resources may be available to survivors in soils with low RS, fueling RL. The fact that we did not observe any consistent tradeoff, especially for N cycling, suggests that resource pools which would be affected by mortality, dormancy, or lysis in the RS response were not major limiting factors to RL.

### 3.3 Relating baseline soil health and microbial communities to C and N cycling resistance and resilience

Stepwise multiple regression was used to identify the baseline soil characteristics most closely related to C and N cycle RS and RL to DRT and D+H stresses (Table 2). The best predictors of RS and RL varied by stress and cycle, with some consistent patterns. We predicted that RS would increase with soil porosity and SOM (hypothesis 2), and that soils with higher chemical fertility will tend to have higher RL (hypothesis 3). In partial support of these hypotheses, "Arableness" (Prin 1; high fertility, low porosity) had significant negative correlations with multifunctional C and N cycle RS to DRT (Table 2), reflecting the FOR site's numerically higher values than the arable sites for both C and N cycle RS to DRT. Conversely, "Arableness" was generally positively correlated to RL, especially for the C cycle in which it was a highly significant positive predictor of RL to both DRT and D+H. Interestingly "Soil health" (Prin 2, which loaded for properties related to microbial biomass, activity, and aggregate stability) was not a good predictor of C-cycle RL (Table 2). This suggests that in the unfertilized, acidic FOR soil, soil chemistry posed a stronger control on community C-cycle RL than factors traditionally considered in a soil health assessment, such as structure and SOM. Soil pH, which was the most influential loading factor on Prin 1 ("Arableness"), has been cited as an important control on multifunctional resistance both on the field [18] and global [14] scales.

Traditional soil health indicators' ("Soil Health"; Prin 2) relationship to RS and RL depended on the cycle and stress. "Soil Health" was positively related to C cycle RS to DRT but slightly negatively to C cycle RS to D+H, and had no relationship to N cycle RS to either stress (Table 2). There were no significant predictors for N-cycle RS to D+H. Conversely, "Soil Health" was a strong positive predictor of N cycle RL to DRT. This result is in line with the hypothesis that in general, ecosystems with a high degree of internal self-regulation will be more buffered to external disturbances [5,9]. Under this theory, cycles which rely more heavily on synthetic inputs, such as N, would have low RS and RL to a stress which affects

**Table 2. Best models identified by stepwise multiple regression using Principal Components (Prin) 1 through 4 as predictors for the multifunctional resistance and resilience of C and N cycling functions. Resistance and Resilience indices were calculated for drought (DRT) and drought+heat (D+H) stressed soils with reference to an unstressed control. P-values are given for every parameter entered into the model and not removed (threshold p<0.15), and for the model as a whole. Signs in parentheses after parameter names represent parameter directionality.**

| Index | Cycle | Stress | Best Predictors | | | | Whole model | |
|---|---|---|---|---|---|---|---|---|
| | | | Parameter | P | Parameter | p | Adj. R² | p |
| Resistance | C | DRT | Prin 2 (+) | **0.043** | Prin 1 (-) | **0.033** | 0.463 | **0.010** |
| | | D+H | Prin 3 (-) | **0.004** | Prin 2 (-) | 0.062 | 0.594 | **0.003** |
| | N | DRT | Prin1 (-) | **0.035** | | | 0.265 | **0.035** |
| | | D+H | Prin1 (+) | 0.131 | | | 0.111 | 0.131 |
| Resilience | C | DRT | Prin1 (+) | **0.007** | | | 0.429 | **0.007** |
| | | D+H | Prin1 (+) | **<0.001** | Prin 2 (+) | 0.091 | 0.692 | **0.001** |
| | N | DRT | Prin 2 (+) | **<0.001** | Prin 3 (-) | **0.037** | 0.789 | **<0.001** |
| | | D+H | Prin1 (+) | 0.072 | | | 0.181 | 0.072 |

the pool or process in question. In our experiment, the synthetically fertilized CONV site with its high residual $NO_3$-N and low microbial biomass, microbial activity and SOM is an example of a system with little power of internal N cycling [35]. The hypothesis that N cycle RL in the CONV system was limited by its low SOM is supported by the higher RL observed under D+H in the CONV system, as the addition of heat stress increased dissolved organic C concentrations [20], which would alleviate this limitation.

We hypothesized that bacterial and fungal diversity would be associated with high RS and RL. However, the principal component relating to community diversity ("Alpha Diversity", Prin 3) was never positively related to RS or RL for either cycle. In fact, it had significant negative relationships with C cycle RS to D+H and N cycle RL to DRT (Table 2). This is in contrast with the results of Zhang et al. [18], who found that bacterial diversity explained more of soil multifunctional RS and RL to a wet-dry stress than did soil properties. Our experimental design did not allow us to isolate the individual effects of different taxa. However, the fact that microbial community composition, soil properties, and RS and RL differed among sites while diversity metrics did not supports the idea that the presence or activity of a particular taxa or edaphic factor was more influential than diversity as such [13].

Generalizations across phyla are not necessarily valid given strong within-phyla diversity [36]. Still, it is interesting to note that the relative abundance of Acidobacteria, many of which are considered to be stress-adapted oligotrophs with high relative abundance under low-SOM conditions [37,38], had a strong negative loading for Prin 2 ("Soil Health"). This phylum also was significantly more abundant in the low-SOM CONV soil than the high-SOM ORG soil. Conversely Firmicutes, which tend to have copiotrophic lifestyles [36], loaded highly on Prin 2 and had the highest relative abundance in the high-SOM ORG and FOR soils. These results suggest that communities adapted to high-SOM conditions may be associated with improved N cycle RL to DRT stress.

## 4. Conclusions

### 4.1 Implications for optimizing land management practices for functional stability under climate variability

The goal of this study was to link soil health status with stress response in three differently managed systems. Our results demonstrate that the long-term practice of management designed to improve soil health appeared to buffer soil RS and RL to climate stress. The ORG soil, which combined long perennial phases with periodic additions of manure and pH adjustment, appears to represent a "middle path" between the conventionally farmed and natural sites. That is, it created an environment characterized by good physical characteristics and high SOM like the natural site, which were associated with greater RS. At the same time, it possessed the adequate chemical fertility and neutral pH of the conventionally farmed site. These resulted in an RS and RL which was always equal to or greater than that of the other two systems, under both stresses. Conversely, conventional practices, even when combined with no-till, led to a less buffered system with relatively low RS to DRT, and very low RL of N cycling processes to DRT. This low N-cycle RL was especially associated with low SOM, biological activity, and porosity, combined with high soil $NO_3$-N and a microbial community which may be more adapted to low-OM conditions.

Despite their wide variation in soil health characteristics, all systems showed poor RS when subjected to heat under drought conditions. An implication is that management which buffers soil temperature, such as the presence of living cover, high-albedo mulch, or crop residues, is a vital component of maintaining functional stability in a warming climate. It is especially important for natural systems to avoid conditions that expose the soil to unaccustomed stresses, as sites with low chemical fertility and an acidic pH may not be able to recover function as quickly.

### 4.2 Implications for soil health assessment

Our results broadly support the assumption that more healthy soils, by traditional agriculturally-focused definitions (e.g., higher SOM and its fractions, microbial activity, and aggregate stability), are more resistant and resilient than less healthy ones. However, our findings highlight the limitations of soil health assessment as applied to natural systems. The "soil

health" concept did not fit the FOR soil, with its high porosity, SOM, and microbial biomass but poor aggregation, high acidity, low fertility, and low enzyme activity. This is reasonable, since soil health is normally assessed in the context of agriculture in which a primary functional goal is sustained crop yields. As noted by Peterson et al. [9], creating a resilient agroecosystem may first require adding the external inputs needed to get it to a productive state (e.g., lime, fertility sources), and then managing so as to maximize its ability to internally self-regulate (e.g., improving SOM, structure, and biological activity). Our natural site's generally good physical characteristics were associated with high RS to DRT for both C and N cycles. However, it had very poor RS when heat was added, likely as a result of a non-adapted microbial community. In addition, its acidity and low chemical fertility were strongly associated with poor stress RL. Our results imply that if the idea of "soil health" for agricultural systems includes the ability to respond well under a stress to which agricultural systems are more subject (such as heat), soil health assessment schemes that propose using metrics from an undisturbed native soil as benchmarks (e.g., [39]) may not be appropriate.

## Supporting information

**S1 File. Do soil health indicators predict carbon and nitrogen functional stability under drought and heat ?**
(DOCX)

**S1 Figa. Bacterial alpha diversity metrics and relative abundance of major phyla.** FOR= Forest soil ORG = soil from a long-term organic annual cropping system CONV = soil from a long-term conventionally-managed cropping system. Different lowercase letters represent statistical difference at p < 0.05 using Tukey HSD (n = 5).
(TIF)

**S1 Figb. Fungal alpha diversity metrics and relative abundance of major phyla.** FOR= Forest soil ORG = soil from a long-term organic annual cropping system CONV = soil from a long-term conventionally-managed cropping system. Different lowercase letters represent statistical difference at p < 0.05 using Tukey HSD (n = 5).
(TIF)

**S1 Table. Mean values of baseline soil properties in all three sites.** BD = Bulk density; GWC = gravimetric water content; WHC = water-holding capacity; WSA = water-stable aggregates; EOC & EON = extractable organic C & N; POXC = permanganate-extractable C; BG, LAP, NAG, XYL = potential activity of β-glucosidase, leucine aminopeptidase, N-acetyl-β-glucosaminidase & xylanase, respectively; MBC &MBN = microbial biomass C & N; Cmin7 & Nmin7 = 7-d C and net N mineralization potential; STBA-24 = soil test biological activity, as measured by 24-h respiration from rewet soil.
(XLSX)

**S2 Table. Loading factors for the first four principal components for baseline soil and microbial variables.** GWC = gravimetric water content; WHC = water-holding capacity; WSA = water-stable aggregates; EOC & EON = extractable organic C & N; POXC = permanganate-extractable C; BG, LAP, NAG, XYL = potential activity of β-glucosidase, leucine aminopeptidase, N-acetyl-β-glucosaminidase & xylanase, respectively; MBC &MBN = microbial biomass C & N; Cmin7 & Nmin7 = 7-d C and net N mineralization potential; STBA-24 = soil test biological activity, as measured by 72-h respiration from rewet soil.
(XLSX)

## Acknowledgments

Thanks to Nathan Hicklin for assistance in site identification and sampling, and to Windy Acres and TC Groves Farms for allowing us to sample their soils. Thanks to Maddie Evans, Michael Evan Russell and Mary Catherine Pile for help in sample analysis.

## Author contributions

**Conceptualization:** Patricia Lazicki, Sindhu Jagadamma.

**Data curation:** Patricia Lazicki.

**Funding acquisition:** Alemu Mengistu.

**Investigation:** Patricia Lazicki.

**Methodology:** Patricia Lazicki.

**Project administration:** Jaehoon Lee, Alemu Mengistu, Sindhu Jagadamma.

**Resources:** Jaehoon Lee, Sindhu Jagadamma.

**Supervision:** Jaehoon Lee, Sindhu Jagadamma.

**Visualization:** Patricia Lazicki.

**Writing – original draft:** Patricia Lazicki.

**Writing – review & editing:** Patricia Lazicki, Jaehoon Lee, Sindhu Jagadamma.

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
