## [Decision Letter · Decision Letter 0]

27 Feb 2025

PONE-D-24-46379Do soil health indicators predict carbon and nitrogen functional stability under drought and heat?PLOS ONE?

Dear Dr. Lazicki,

Thank you for submitting your manuscript to PLOS ONE. After careful consideration, we feel that it has merit but does not fully meet PLOS ONE’s publication criteria as it currently stands. Therefore, we invite you to submit a revised version of the manuscript that addresses the points raised during the review process.

We look forward to receiving your revised manuscript.

Kind regards,

Dafeng Hui, Ph.D.

Academic Editor

PLOS ONE

Journal Requirements:

3. We noted in your submission details that a portion of your manuscript may have been presented or published elsewhere. Please clarify whether this [conference proceeding or publication] was peer-reviewed and formally published. If this work was previously peer-reviewed and published, in the cover letter please provide the reason that this work does not constitute dual publication and should be included in the current manuscript.

4. In the online submission form, you indicated that your data will be submitted to a repository upon acceptance.  We strongly recommend all authors deposit their data before acceptance, as the process can be lengthy and hold up publication timelines. Please note that, though access restrictions are acceptable now, your entire minimal  dataset will need to be made freely accessible if your manuscript is accepted for publication. This policy applies to all data except where public deposition would breach compliance with the protocol approved by your research ethics board. If you are unable to adhere to our open data policy, please kindly revise your statement to explain your reasoning and we will seek the editor's input on an exemption.

“USDA ARS (grant number 58-6066-8-043); Tennessee Department of Agriculture (grant number A20-0337)”

“USDA ARS (grant number 58-6066-8-043); Tennessee Department of Agriculture (grant number A20-0337)”

Additional Editor Comments (if provided):

I now have two reports on the manuscript from expert reviewers. Both reviewers were positive, but also raised some technical concerns on the current version of the manuscript. The authors need to make substantial revisions before it can be accepted for publication.

Reviewers' comments:

Reviewer's Responses to Questions

**Comments to the Author**

1. Is the manuscript technically sound, and do the data support the conclusions?

Reviewer #1: Yes

Reviewer #2: Yes

2. Has the statistical analysis been performed appropriately and rigorously?

Reviewer #1: Yes

Reviewer #2: Yes

3. Have the authors made all data underlying the findings in their manuscript fully available?

Reviewer #1: Yes

Reviewer #2: Yes

4. Is the manuscript presented in an intelligible fashion and written in standard English?

Reviewer #1: Yes

Reviewer #2: Yes

Reviewer #1: The manuscript titled "Do soil health indicators predict carbon and nitrogen functional stability under drought and heat?" delves into a critical and contemporary issue: the intricate relationship between soil health indicators and the functional stability of soil carbon (C) and nitrogen (N) cycling in the face of climate-induced stress. Unraveling how soil health modulates resistance and resilience to drought and heat is paramount for sustainable soil management and effective climate change adaptation. This study offers profound insights into the differential impacts of various management practices—forest, conventional, and organic—on soil functional stability. The strategic selection of three distinct land-use types serves as an ideal framework for comparing soil health and functional stability across diverse agricultural approaches. The sampling methodology, which encompasses the collection of undisturbed soil cores and bulk samples, is meticulously crafted to encapsulate both spatial and functional variability.

The experimental setup, encompassing both drought and combined drought-heat stress scenarios, is pertinent to projected climate change models. The 28-day stress and recovery period is adequately designed to capture significant alterations in soil functions. Utilizing a multifunctional approach to evaluate carbon and nitrogen cycling provides a robust assessment of soil functional stability. Employing 16S and ITS sequencing for analyzing microbial community diversity and composition offers a rigorous methodology to elucidate the role of microbial communities in maintaining soil function. The comprehensive inclusion of both bacterial and fungal communities aligns with contemporary understanding in soil microbiology.

The findings that microbial community composition exhibits significant variation among different sites, while diversity metrics remain relatively consistent, suggest that specific taxa or community structure may play a more crucial role than overall diversity in determining functional stability. Identifying key soil properties such as pH levels, chemical fertility, and porosity that differentiate these sites is essential for elucidating the underlying mechanisms governing soil functional stability.

The results indicate that resistance (RS) and resilience (RL) exhibit variability depending on the type of stressor and land use practices, underscoring the intricate nature of soil functional stability. Notably, the findings reveal that the addition of heat to drought conditions significantly diminishes RS and RL, a critical insight for regions subjected to multiple climate stressors. Furthermore, the observation that organic management practices generally promote higher RL compared to conventional and forested sites implies that such practices may play a pivotal role in enhancing soil functional stability.

The application of principal components analysis (PCA) and stepwise multiple regression to identify predictors of RS and RL is statistically robust. The discovery that soil health indicators, such as total organic carbon and microbial biomass, are correlated with nitrogen cycle resilience under drought stress, is crucial for the development of comprehensive soil health assessment frameworks. The absence of a consistent relationship between microbial diversity and functional stability implies that other factors, including community composition and soil properties, may exert a more substantial influence.

The study concludes that conventional soil health indicators, as traditionally defined within agricultural contexts, may not comprehensively capture the functional stability of natural ecosystems. The organic management system appears to effectively balance the advantages of high soil organic matter and microbial activity with sufficient chemical fertility, thereby enhancing functional stability. This conclusion is robustly supported by empirical data and offers practical guidance for sustainable soil management practices.

In Section 2.4.3, the equations for calculating RS and RL are clearly presented; however, incorporating additional explanatory text would enhance readers' comprehension of the rationale underpinning these calculations.

Several sections include redundant information. For instance, the analysis of microbial community diversity and its correlation with functional stability is addressed repeatedly across multiple sections. To enhance clarity and conciseness, this information should be consolidated.

Some sentences are complex and could be simplified for clarity. For example, in Section 3.3, the sentence "This suggests that in the unfertilized, acidic FOR soil, soil chemistry posed a stronger control on community RL to DRT than factors traditionally considered in a soil health assessment" This sentence could be divided into two sentences for improved readability.

The manuscript is scientifically rigorous and offers valuable insights into the relationship between soil health indicators and functional stability under climate stress. The experimental design is robust, and the results are thoroughly supported by comprehensive data. While the manuscript is well-written, there are minor grammatical and stylistic enhancements that could further improve clarity. This study makes a significant contribution to the fields of soil science and sustainable agriculture.

Reviewer #2: This study provides valuable insights into the complex relationships between soil health indicators and functional resilience under climate stress. By linking carbon and nitrogen cycling responses to different management practices, it highlights key factors influencing soil resistance and resilience. The findings offer a strong scientific basis for improving soil management strategies under climate change. However few points are to be properly addressed for the clarity of the readers (appended below):

1. The introduction mentions microbiota but could briefly specify key microbial groups (bacteria, fungi, archaea) that drive carbon and nitrogen cycling.

2. The explanation of RS and RL should be refined for clarity. Instead of "more quickly and completely to regain function," use "recover function more rapidly and effectively."

3. While the introduction discusses soil organic carbon (SOC), porosity, and fertility, it should explicitly state which soil health assessment methods are commonly used and how they may overlook stress response.

4. The transition to multi-functionality indices should be smoother, explaining their relevance in quantifying overall ecosystem function rather than listing citations.

5. The hypotheses should be more precise:

- Specify how high porosity and SOC reduce water loss and heat transmission.

- Explain why microbial diversity may enhance both RS and RL by fostering functional redundancy.

6. The methodology section should be more structured by separating different processes with clear subheadings. Consider breaking down complex sentences for readability.

7. Ensure uniformity in naming conventions, such as consistently referring to "CONV," "ORG," and "FOR" throughout the text without alternation between abbreviations and full forms.

8. In certain methods, like enzyme activity assays and microbial community analysis, Briefly summarizing key steps within the main text will enhance clarity.

9. Ensure uniformity in how measurements are reported (e.g., using consistent units for soil moisture and enzyme activities). Some values, such as temperatures and durations, should be explicitly stated in standard scientific formats.

10. How do microbial community composition and diversity vary across forest, organic and conventional land-use systems?

11. What are the key soil physicochemical properties driving microbial community shifts under different land-use types.

12. How do carbon and nitrogen cycling processes respond to single and combined drought-heat stress across distinct land-use systems.

13. What mechanisms underpin microbial functional resilience and resistance to environmental stressors in different land-use regimes?

14. How can land management practices be optimized to enhance microbial stability and ecosystem functions under climate variability?

15. How can soil health assessment frameworks be modified to better account for the unique resilience and stress responses of natural ecosystems, rather than relying solely on agricultural benchmarks?

16. What specific microbial adaptations contribute to improved resistance and resilience under heat stress, and how can these be leveraged to enhance soil health in both natural and managed systems?

I hope that the authors shall go through the points and revise the MS accordingly for its possible publication.

**Do you want your identity to be public for this peer review?** For information about this choice, including consent withdrawal, please see our Privacy Policy

Reviewer #1: No

Reviewer #2: No

---

## [Author Response · Author response to Decision Letter 1]

31 Mar 2025

Response to reviewers

Reviewer #1: The manuscript titled "Do soil health indicators predict carbon and nitrogen functional stability under drought and heat?" delves into a critical and contemporary issue: the intricate relationship between soil health indicators and the functional stability of soil carbon (C) and nitrogen (N) cycling in the face of climate-induced stress. Unraveling how soil health modulates resistance and resilience to drought and heat is paramount for sustainable soil management and effective climate change adaptation. This study offers profound insights into the differential impacts of various management practices—forest, conventional, and organic—on soil functional stability. The strategic selection of three distinct land-use types serves as an ideal framework for comparing soil health and functional stability across diverse agricultural approaches. The sampling methodology, which encompasses the collection of undisturbed soil cores and bulk samples, is meticulously crafted to encapsulate both spatial and functional variability.

The experimental setup, encompassing both drought and combined drought-heat stress scenarios, is pertinent to projected climate change models. The 28-day stress and recovery period is adequately designed to capture significant alterations in soil functions. Utilizing a multifunctional approach to evaluate carbon and nitrogen cycling provides a robust assessment of soil functional stability. Employing 16S and ITS sequencing for analyzing microbial community diversity and composition offers a rigorous methodology to elucidate the role of microbial communities in maintaining soil function. The comprehensive inclusion of both bacterial and fungal communities aligns with contemporary understanding in soil microbiology.

The findings that microbial community composition exhibits significant variation among different sites, while diversity metrics remain relatively consistent, suggest that specific taxa or community structure may play a more crucial role than overall diversity in determining functional stability. Identifying key soil properties such as pH levels, chemical fertility, and porosity that differentiate these sites is essential for elucidating the underlying mechanisms governing soil functional stability.

The results indicate that resistance (RS) and resilience (RL) exhibit variability depending on the type of stressor and land use practices, underscoring the intricate nature of soil functional stability. Notably, the findings reveal that the addition of heat to drought conditions significantly diminishes RS and RL, a critical insight for regions subjected to multiple climate stressors. Furthermore, the observation that organic management practices generally promote higher RL compared to conventional and forested sites implies that such practices may play a pivotal role in enhancing soil functional stability.

The application of principal components analysis (PCA) and stepwise multiple regression to identify predictors of RS and RL is statistically robust. The discovery that soil health indicators, such as total organic carbon and microbial biomass, are correlated with nitrogen cycle resilience under drought stress, is crucial for the development of comprehensive soil health assessment frameworks. The absence of a consistent relationship between microbial diversity and functional stability implies that other factors, including community composition and soil properties, may exert a more substantial influence.

The study concludes that conventional soil health indicators, as traditionally defined within agricultural contexts, may not comprehensively capture the functional stability of natural ecosystems. The organic management system appears to effectively balance the advantages of high soil organic matter and microbial activity with sufficient chemical fertility, thereby enhancing functional stability. This conclusion is robustly supported by empirical data and offers practical guidance for sustainable soil management practices.

1. In Section 2.4.3, the equations for calculating RS and RL are clearly presented; however, incorporating additional explanatory text would enhance readers' comprehension of the rationale underpinning these calculations.

This is an excellent point. We have added some text to the paragraph in question, explaining some of the rationale. The introduction to the equations now reads (lines 203-207):

“Resistance (RS) and resilience (RL) indices were calculated for each function using the method of Orwin and Wardle [22]. This method is commonly used in microbial response studies as it, a) increases monotonically as RS or RL increase, b) gives equivalent values for a response of the same magnitude, whether it is higher or lower than that of the control, c) avoids infinite values and zero denominators and d) standardizes the recovery value to the initial disturbance [22].”

2. Several sections include redundant information. For instance, the analysis of microbial community diversity and its correlation with functional stability is addressed repeatedly across multiple sections. To enhance clarity and conciseness, this information should be consolidated.

We have gone through the MS and looked for redundant statements so that they can be consolidated.

3. Some sentences are complex and could be simplified for clarity. For example, in Section 3.3, the sentence "This suggests that in the unfertilized, acidic FOR soil, soil chemistry posed a stronger control on community RL to DRT than factors traditionally considered in a soil health assessment" This sentence could be divided into two sentences for improved readability.

The manuscript is scientifically rigorous and offers valuable insights into the relationship between soil health indicators and functional stability under climate stress. The experimental design is robust, and the results are thoroughly supported by comprehensive data. While the manuscript is well-written, there are minor grammatical and stylistic enhancements that could further improve clarity. This study makes a significant contribution to the fields of soil science and sustainable agriculture.

We have gone through the MS and looked for places where sentences can be simplified without losing their logical flow, and where grammar can be improved. Examples include Lines 125-128, Lines 141-144, Lines 162-164, Lines 172-174, Lines 182-186

Reviewer #2: This study provides valuable insights into the complex relationships between soil health indicators and functional resilience under climate stress. By linking carbon and nitrogen cycling responses to different management practices, it highlights key factors influencing soil resistance and resilience. The findings offer a strong scientific basis for improving soil management strategies under climate change. However few points are to be properly addressed for the clarity of the readers (appended below):

1. The introduction mentions microbiota but could briefly specify key microbial groups (bacteria, fungi, archaea) that drive carbon and nitrogen cycling.

Addressed in line 45-46. The sentence now reads:

“These extreme climate events can all be stressors on the bacteria, fungi, and archaea that drive important soil functions such as carbon (C) and nitrogen (N) cycling [1,2].”

2. The explanation of RS and RL should be refined for clarity. Instead of "more quickly and completely to regain function," use "recover function more rapidly and effectively."

Addressed in line 50-53 with a slight change; it’s important to note that resemblance to non-stressed conditions is an important component of RS, and this change is defined both in terms of how long it takes, and how far the stressed and non-stressed values are from each other.:

“That is, important microbially-driven functions in “healthy” soils are assumed to change less during stress (RS) and to more rapidly and closely resemble those of non-stressed soils when conditions return to normal (RL) [3].”

3. While the introduction discusses soil organic carbon (SOC), porosity, and fertility, it should explicitly state which soil health assessment methods are commonly used and how they may overlook stress response.

We have added a sentence to clarify broadly the types of soil health metrics commonly used (e.g. various pools and processes—there are a lot of soil health assessment schemes, and it would take too long to list them all), and that these schemes don’t measure how these pools or processes respond to climate stress. (Lines 55-58)

“However, soil health assessment schemes, which normally include soil organic matter and its various fractions, available nutrients, physical properties, and indices of soil biological activity such as respiration or potential enzyme activities, generally do not include how these pools and functions respond to climate stress (e.g. [6,7]).”

4. The transition to multi-functionality indices should be smoother, explaining their relevance in quantifying overall ecosystem function rather than listing citations.

Excellent point! We have added some more explanation on the benefits of multifunctionality indices (Lines 77—81:

“This approach has long been used in ecology to study the links between organisms and ecosystem function, as it provides a means of summarizing the aggregate functional change of an ecosystem in response to a change that simultaneously affects all members of the community [16]. More recently, the approach has been combined with RS and RL indices to relate the stress behavior of soil processes to soil and microbial community characteristics [17,18].”

5. The hypotheses should be more precise:

- Specify how high porosity and SOC reduce water loss and heat transmission.

- Explain why microbial diversity may enhance both RS and RL by fostering functional redundancy.

We have made the suggested changes. The hypotheses now read (Lines 90-96):

“1) Adding heat to drought would decrease RS and RL of all measured C and N cycle metrics at all sites [11], 2) RS would increase with soil porosity and soil C, as such soils will likely lose water and transmit heat more slowly due to better water holding capacity and improved thermal resistivity [10,19], 3) High chemical fertility would be associated with high RL, as more available nutrients will support faster microbial regrowth [8], and 4) Higher bacterial and fungal diversity would be associated with high RS and RL, as increased functional redundancy increases the likelihood of organism’s ability to perform any given function under both stress and recovery conditions [18].”

6. The methodology section should be more structured by separating different processes with clear subheadings. Consider breaking down complex sentences for readability.

We carefully reread the methods and broke up some of the longer sentences. Examples include Lines 125-128, Lines 141-144, Lines 162-164, Lines 172-174, Lines 182-186

If we understand the comment correctly, the reviewer suggests grouping C and N cycling processes separately. While this suggestion is reasonable, the nature of the experiment is to measure the change in many processes simultaneously in response to an imposed stress and recovery. Therefore, we think that it would be more difficult to describe the experiment if we structured by separate processes rather than by chronology of laboratory steps (e.g. baseline soil properties, imposition and lifting of stress, and various functional measurements performed at each timepoint), as it is currently organized. Additionally, separating C and N cycling functions would greatly lengthen the manuscript as it would involve describing or at least referencing the same experiment twice where both C and N cycling functions were measured. We think this would also reduce clarity as it might be less evident that many functions (e.g. C and N mineralization, MBC, MBN, C- and N-cycling activities) were all measured on the same physical soil core in the course of the same experiment.

To improve the clarity of which functions were measured when, we have re-ordered so the paragraphs describing which functions were designated as either C or N -cycling are located directly after the description of the incubation experiment, under the subhead "C and N cycling functions”. We also added some language to make it clearer on which fractions the cores were measured. Unfortunately, due to journal style rules, we could not add an additional subhead for each paragraph, but we added language to make it immediately clear whether it referred to C or N cycling. We hope that these changes address the reviewer’s concerns. The paragraphs read (lines 187-200)

“Functions were designated as related to C or N cycling. The nine N cycling functions were a) From the undisturbed cores: mineral N, the proportion of the mineral N measured which was in the form of NH4, MBN, EON, and the potential activity of the enzymes leucine aminopeptidase (LAP) and N-acetyl-β-glucosaminidase (NAG) from undisturbed cores; b) From unamended, sieved soil during incubation: net N mineralized over 7 d and c) From residue-amended soil after 7 d incubation: additional mineral N and MBN measured, compared to unamended soils.

The ten C cycling functions were: a) From the undisturbed cores: cumulative total respiration per unit MBC, MBC, EOC, and the potential activity of the enzymes β-glucosidase (BG) and xylanase (XYL) b) From unamended soil during incubation: CO2-C mineralized over 24 h and 7 d, CO2-C mineralized over 24 h per unit MBC (qCO2), and c) From residue-amended soil over 7 d incubation: additional MBC and CO2-C measured, compared to unamended soils.”

7. Ensure uniformity in naming conventions, such as consistently referring to "CONV," "ORG," and "FOR" throughout the text without alternation between abbreviations and full forms.

We have done a text search throughout the MS and eliminated any alternation. However, we have kept the unabbreviated form when used in reference to something other than our experimental soil descriptors (e.g. organic matter rather than ORG matter). We have similarly corrected some inconsistencies in SOM/SOC.

8. In certain methods, like enzyme activity assays and microbial community analysis, Briefly summarizing key steps within the main text will enhance clarity.

Because of the very large number of analyses, and because we followed standard methods, we elected to describe our sampling and storage procedures but referred the reader to the standard method for more analytical details. Otherwise, the methods section would be prohibitively long. We have included a detailed method for DNA extraction, community analysis, and enzyme analysis in the Supplemental methods. If the reviewer strongly feels that there are more details that it’s important to include in the main text we can consider adding these. At the moment, we added some wording to briefly orient the reader to the specific method used (lines 128-133)

“Subsamples were stored at -80°C for microbial community quantification via 16s and ITS analyses and at -20 °C for enzyme analysis. Soil microbial DNA was extracted from five 0.25-g replicate subsamples of the frozen composite sample collected at each site using the Qiagen DNeasy PowerSoil kit, according to the manufacturer's instructions (Qiagen, Valencia, CA, United States). Enzyme analyses were performed using the fluorescence microplate method (Table 1). Details of DNA extraction, community analysis, and enzyme assays are given in the Supplemental Methods.”

9. Ensure uniformity in how measurements are reported (e.g., using consistent units for soil moisture and enzyme activities). Some values, such as temperatures and durations, should be explicitly stated in standard scientific formats.

Temperatures are stated in °C , durations as h or d when used as a unit, or “day” if referring to a particular sampling day, e.g. ”on day 28”. Moisture content has been changed from “g H2O g-1 soil” to “a water content of X g g-1 dry soil”. Enzyme measurements are not reported. A text search was performed to ensure consistency in measurements.

10. How do microbial community composition and diversity vary across forest, organic and conventional land-use systems?

We have included this information in the p

---

## [Decision Letter · Decision Letter 1]

7 May 2025

Do soil health indicators predict carbon and nitrogen functional stability under drought and heat?

PONE-D-24-46379R1

Dear Dr. Lazicki,

We’re pleased to inform you that your manuscript has been judged scientifically suitable for publication and will be formally accepted for publication once it meets all outstanding technical requirements.

Kind regards,

Dafeng Hui, Ph.D.

Academic Editor

PLOS ONE

Additional Editor Comments (optional):

Reviewers' comments:

Reviewer's Responses to Questions

**Comments to the Author**

Reviewer #1: All comments have been addressed

Reviewer #2: All comments have been addressed

2. Is the manuscript technically sound, and do the data support the conclusions?

Reviewer #1: Yes

Reviewer #2: Yes

3. Has the statistical analysis been performed appropriately and rigorously?

Reviewer #1: Yes

Reviewer #2: Yes

4. Have the authors made all data underlying the findings in their manuscript fully available?

Reviewer #1: Yes

Reviewer #2: Yes

5. Is the manuscript presented in an intelligible fashion and written in standard English?

Reviewer #1: Yes

Reviewer #2: Yes

Reviewer #1: The authors have conducted a comprehensive revision of the manuscript, substantially improving its overall quality.

Reviewer #2: Dear Authors, I am happy to see the changes made in the revised manuscript. All the comments were taken care off and this MS now can be accepted on the discretion of EIC / AE.

**Do you want your identity to be public for this peer review?** For information about this choice, including consent withdrawal, please see our Privacy Policy

Reviewer #1: **Yes: ** Junqiang Zheng

Reviewer #2: No

---

## [Editor Report · Acceptance letter]

PONE-D-24-46379R1

PLOS ONE

Dear Dr. Lazicki,

I'm pleased to inform you that your manuscript has been deemed suitable for publication in PLOS ONE. Congratulations! Your manuscript is now being handed over to our production team.

Kind regards,

on behalf of

Dr. Dafeng Hui

Academic Editor

PLOS ONE